# Comparative Study of Plywood Boards Produced with Castor Oil-Based Polyurethane and Phenol-Formaldehyde Using *Pinus taeda* L. Veneers Treated with Chromated Copper Arsenate

Estefani Sugahara [1], Bruno Casagrande [1], Felipe Arroyo [2], Victor De Araujo [2], Herisson Santos [3], Emerson Faustino [3], Andre Christoforo [2,*] and Cristiane Campos [1,*]

1   Science and Engineering Institute, São Paulo State University, Itapeva 18409-010, Brazil;
    estefani.sugahara@unesp.br (E.S.); bcasagrande@outlook.com.br (B.C.)
2   Exact Sciences and Technology Center, Federal University of São Carlos, São Carlos 13565-905, Brazil;
    lipe.arroyo@gmail.com (F.A.); engim.victor@yahoo.de (V.D.A.)
3   Campus of Ariquemes, Federal Institute of Education, Science & Technology of Rondônia,
    Ariquemes 76870-000, Brazil; herisson.santos@ifro.edu.br (H.S.); emerson.faustino@ifro.edu.br (E.F.)
*   Correspondence: alchristoforo@ufscar.br (A.C.); cristiane.campos@unesp.br (C.C.)

**Abstract:** Plywood is widely used in civil construction. Due to the importance of preservation and gluing in bio-composites, this study compares the influence of a chemical treatment with CCA (chromated-copper-arsenate) on *Pinus taeda* L. wood veneers to produce two plywood types using phenol-formaldehyde (PF) and castor oil-based polyurethane (PU). Four different treatments were performed to analyze both varieties' physical and mechanical properties. As a result, an improvement in the properties of the treated panels was observed. Lower moisture contents and better interactions caused by less thickness swelling and water absorption were identified in the PU-based plywoods. The treatment with CCA was efficient, improving these properties when they were compared to the reference panels. Most treatments evidenced increases in the modulus of elasticity and modulus of rupture for both adhesives when the CCA treatment was applied to the veneers. Comparing the resins, the PF showed the best values of modulus of elasticity. All treatments met the requirements defined by the Brazilian standard document for the glue line shear.

**Keywords:** chromated-copper-arsenate; *Pinus taeda*; bio-composite; phenol-formaldehyde; castor oil-based polyuretane; physical and mechanical properties





## 1. Introduction

Wood is often described as being essential for mankind, due its large quantity, high renewability and environmentally benign qualities, ease of working and excellent mechanical properties [1,2]. The wood industry has grown across the world because it offers a renewable raw material that is increasingly available on the market. Additionally, Brazil is between the countries with the most developed manufacturing processes for wood-based panels from reforestation trees [3]. Pines and eucalypts have been produced from extensive planted forests to supply different sectors of the Brazilian timber industry [4]. Today, four pine essences (Slash pine, Caribbean pine, Ocote pine, and Loblolly pine) are among the main wood species applied to added-value construction parts and panels in Brazil [5]. Pine is used to produce lumber, plywoods and other value-added goods [4].

The global demand for wood-based products has been growing in recent years [6,7]. For example, the consumption of plywood increased globally by around 2.5% from 2015 to 2019, since it reached 108 million m$^3$ in 2019 for a production of only 107 million m$^3$. In this period, the consumption increased to the point of exceeding its production [7].

Plywood is a wood-based panel produced from veneer sheets in its standard model, although other variations can use laths, strips, blocks or battens in their central layers [7].

Plywood has been incorporated into the class of products that support construction industrialization, as it optimizes resources by reducing costs and minimizing waste [8].

Plywood has been preferentially consumed by the furniture industry in Brazil, while a new market niche is being created by lightweight construction manufacturers, due to the greater uses of the structural version of this panel [4]. In practice, this panel may be structurally applied for closing wall surfaces [9,10] and floor applications [11]. Plywood has become the second most popular choice in the Brazilian timber housing sector, followed by oriented strand boards [10]. In addition to construction, plywoods have been widely applied for furniture [4,9].

However, for its safe use in construction systems in the tropical areas such as Brazil, plywood needs to be subjected to treatments to preserve and stabilize it, as this panel is more susceptible to attacks of xylophagous organisms and the effects of temperature and moisture [9]. Consequently, wood protection has become an essential research field. Commonly, wood can be treated by impregnation methods, such as creosote treatment, and surface alteration, such as chromated-copper-arsenate, (CCA) or copper salt treatments [2].

Phenolic resins are acknowledged as the oldest synthetic thermosetting polymers [12]. The formaldehyde-based adhesives normally demonstrate high bond strength and still dominate the wood-based panel industry, especially given their low costs [13,14]. However, due the concerns about the potential indoor pollution, the development of environmentally friendly adhesives that are nontoxic and renewable is meaningful [12,13]. Along with chemical versatility, high reactivity and excellent adhesive performance, the formaldehyde-based resins are related to some problems, such as free formaldehyde in the adhesives and the formaldehyde emission from the wood composites, as the volatile organic compounds are carcinogenic to humans and harmful to the environment [15]. In order to control pollution as well as to minimize energy resources, formaldehyde resins are poorer alternatives, since they are toxic and the raw materials needed to synthesize them come from non-renewable petroleum-based resources.

Wood industry has been attracted by numerous studies about possible alternatives on the utilization of ecofriendly wood adhesives [12,16–19]. For this, tests with different options are so important for the development of wood-based boards, including finding choices based on different resins and treatments to maintain the characteristics of these composites with a lower impact on the environment and humans.

In the field of panel production, researchers have developed studies aiming to enhance the durability and obtain better product properties. Thus, the wood modification is a significant area of study, given the benefits of reducing wood swelling, microbial decay and insect attack [2]. Furthermore, the investigation into the sustainable application of renewable resources is crucial for the world, due to growing anthropic activities causing environmental disorders, i.e., climate change, and soil and air pollution.

In this way, studies have been carried out on the development, evaluation or comparison of different adhesives and treatments to understand the behavior, impact and performance on wood products [16]. Frihart et al. [2] compared the bonding capacity of four adhesives in wood that was chemically modified. Li et al. [16] evaluated a novel environment-friendly adhesive based on recycling of *Broussonetia papyrifera* (L.) L'Her. ex Vent. leaf waste protein. Aristri et al. [20] studied a bio-based polyurethane resin from tannin. Cai et al. [21] prepared a high-performance resin from soy protein isolate and hybrid biomaterial to replace harmful formaldehyde adhesives for plywood production. Zhang et al. [13] tested a tough, water-resistant, high bond strength resin derived from soybean meal and flexible hyper-branched aminated starch for plywood. Peng et al. [22] designed a new method by using tannin for partial substitution of urea formaldehyde resin and inserted plasma pretreatment of wood to strengthen the bonding performance of plywood.

Huzyan et al. [12] studied ecofriendly wood adhesives from date palm fronds lignin for plywood. Ferreira et al. [9] analyzed the static bending strength of heat-treated and chromated copper arsenate-treated plywood. Bekhta et al. [23] studied the effect of heat

treatment on some physical and mechanical properties of birch plywood. Hsu et al. [24] verified the physical-mechanical properties and creep behavior of plywood composed of fully and partially heat-treated veneers.

In this context and relevance for the wood industry, the present study aimed to compare the influence of chemical treatment with and without CCA on the physical and mechanical properties of plywood produced with Loblolly pine wood veneers and two different adhesives, phenol-formaldehyde (PF) and castor oil polyurethane (PU).

However, our material selections were based on the following justifications:

- Loblolly pine (*Pinus taeda* Linnaeus) wood was considered due to its wide availability and utilization in Brazil, as cited by De Araujo et al. [5], for the timber industry and construction applications.
- Both adhesives were considered due to their easy commercial availability worldwide.
- CCA preservative was considered due to the greater aggressiveness of this chemical compound in protecting against wood degradation. Despite some restrictions and prohibitions on the use of CCA in different countries of the Northern hemisphere, the warmer climate and the greater proliferation of wood-decaying insects in the Southern Hemisphere region represent complex obstacles that still justify the use of this more powerful preservative to preserve wood-based parts and panels.

## 2. Materials and Methods

### 2.1. Materials

To produce plywood, Loblolly pine (*Pinus taeda* Linnaeus) wood veneers were used through the donation from the *Caribea Compensados* company, São Manuel, Brazil. These bioresources presented 400 mm × 400 mm × 2.3 mm nominal dimensions, whose veneers were stabilized at 3% moisture content, as performed by Ferreira et al. [9]. The veneers were identified as resistance classes II, III and IV (Figure 1) through the classification of the ABNT ISO 2426-3 [25], in accordance with the intrinsic characteristics of the wood.

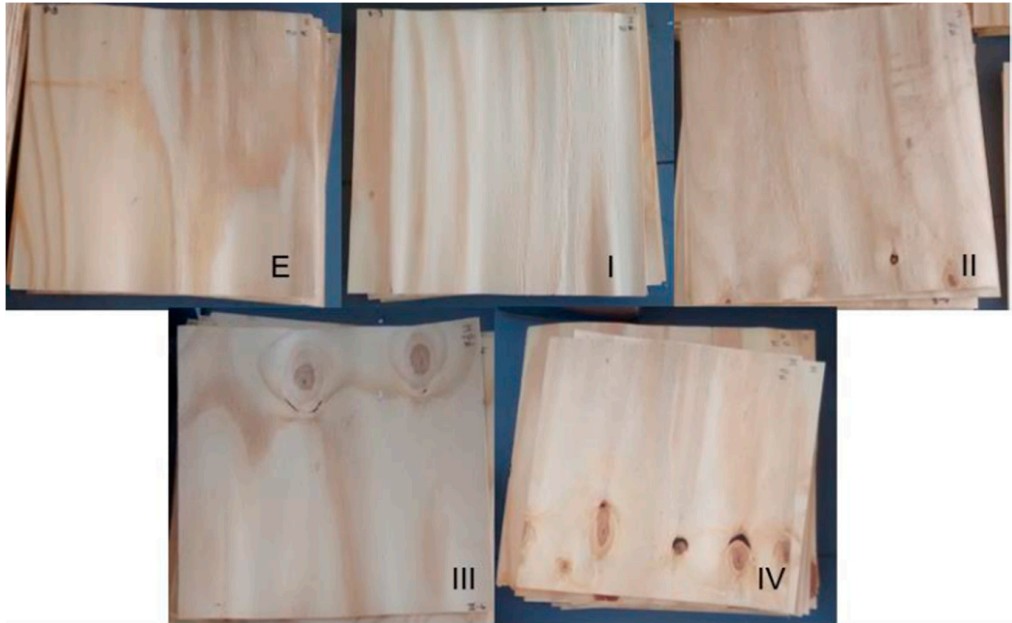

**Figure 1.** Veneer sorting by surface appearance using prescriptions from the ABNT ISO 2426-3: (E) no defects, (I) minimum defect, (II) isolated defects, (III), few defects, and (IV) several defects.

The first adhesive was prepared with a blend of phenol-formaldehyde resin (62% solids content, pH 11, and viscosity of 415 cP), wheat flour and water in a 100:10:10 ratio. The second adhesive was produced using two-component polyurethane resin derived from castor oil beans (100% solids content) in a 1:1 ratio.

The veneer treatment was carried out with chromated-copper-arsenate (CCA type C, composed of 47.5% $CrO_3$, 18.5% $CuO$, and 34% $As_2O_5$, with a solution concentration of 1.6%) by the full cell method in an autoclave.

### 2.2. Methods

#### 2.2.1. Preservative Treatment with CCA

The chromated copper arsenate (CCA) treatment was applied prior to the production of plywood panels for both types of adhesives. The veneers were tied (Figure 2a) and the full cell method in an autoclave was used, consisting of three steps. The first stage consisted of an initial vacuum of 560 mmHg in order to remove the air still present in the pores of the veneers, facilitating the CCA penetration. After 30 min, the autoclave was flooded with the preservative. Then, in the second stage, 9 kgf/cm$^2$ was applied for 60 min, during which time the CCA solution penetrated the wood.

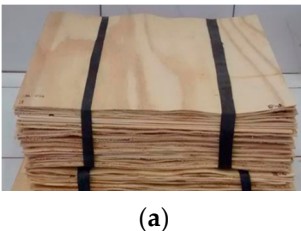 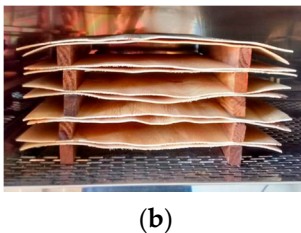 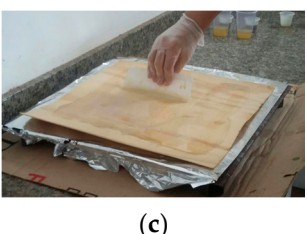

(**a**) (**b**) (**c**)

**Figure 2.** Veneers: (**a**) tying, (**b**) drying, and (**c**) gluing processes.

In the last and third stage, there was a final vacuum of 560 mmHg for 15 min, removing the excess solution from the surface of the wood. After autoclave preservation, the veneers were removed and dried outdoors until they reached a moisture equilibrium content (i.e., approximately 12%).

#### 2.2.2. Plywood Production

Plywoods are manufactured from veneers bonded in the direction of the grain in alternate plies usually orthogonal to each other [7]. From this well-known fundamental, wooden veneers were placed and dried in an oven at $105 \pm 2\ ^\circ$C until they reached a moisture content of 3% to produce the panels (Figure 2b). They were visually classified and only free-defect veneers—that is, without natural knots and cracks—were selected. Four different conditions were analyzed (Table 1). Four panels were made per treatment.

**Table 1.** Treatments analyzed.

| Treatment | Preservative | Adhesive |
|:---:|:---:|:---:|
| T1 | No treated (Reference) | PF [2] |
| T2 | No treated (Reference) | PU [3] |
| T3 | CCA [1] | PF |
| T4 | CCA | PU |

[1] Chromated-copper-arsenate. [2] Phenol-formaldehyde. [3] Castor oil-based polyurethane.

Five-layer plywood was prepared in the experiment by evenly coating the resin on one side of the veneers and overlapping the veneers one with the other, always alternating the orthogonal position. The two resins were applied manually using a plastic spreader, seeking uniformity in the distribution of the amount of adhesive on each veneer, applied in the weight of 395 g/m$^2$ (per panel) in a double glue line (Figure 2c).

After assembly, using a procedure similar to Ferreira et al. [9], the panels were initially cold pre-pressed in a manual press with 1 kgf/cm$^2$ for 20 min to evenly spread the resin and remove excess of air, preventing the bubble formation.

Subsequently, the veneers were hot-pressed with 6 kgf/cm$^2$ at 180 $^\circ$C for 600 s, divided into three pressing cycles with 3 min each and 30 s of pressure relief between cycles and

resulting in plywood panels with a nominal thickness of approximately 11.5 mm [9]. The selected hot-pressing regime was adopted through two depressurized intervals with 30 s to eliminate the vapor generated during the heating transfer.

### 2.2.3. Physical and Mechanical Properties Evaluation

Physical and mechanical tests were performed in accordance with the procedures of different standard documents prescribed by the ABNT (Brazilian Association of Technical Standards), as demonstrated by Table 2.

**Table 2.** Physical and mechanical evaluations.

| Property | Standard |
| --- | --- |
| Specific apparent mass | ABNT NBR 9485:2011 [26] |
| Moisture content | ABNT NBR 9484:2011 [27] |
| Thickness swelling | ABNT NBR 9535:2011 [28] |
| Water absorption | ABNT NBR 9486:2011 [29] |
| Glue line shear | ABNT NBR ISO 12466-1:2012 [30] |
| Parallel and perpendicular static bending | ABNT NBR 9533:2013 [31] |

The Figure 3 presents the samples' dimensions to characterize the panels for each property, including the specific apparent mass and moisture content, thickness swelling, water absorption, glue line shear and parallel, and perpendicular static bending.

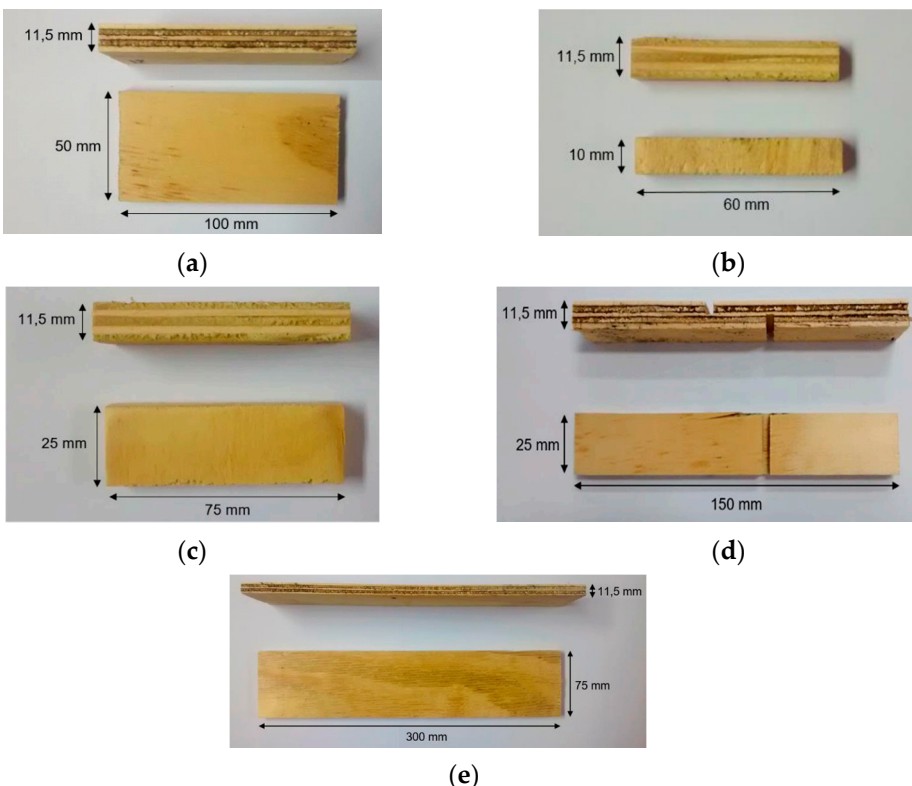

**Figure 3.** Samples' dimensions used to characterize the panels: (**a**) moisture content, (**b**) thickness swelling, (**c**) water absorption, (**d**) glue line shear and parallel and (**e**) perpendicular static bending.

Ten specimens were used per physical test, and six specimens per mechanical test. The results were submitted to analysis of variance and the Tukey test at a significance level of 5% using software R version 3.0 (R Foundation, Auckland, New Zealand). It is possible to identify the factor levels with statistically equivalent means, as well as the levels associated with the lowest and highest mean value. It is worth noting that, from the Tukey

test, 'a' denotes the treatment associated with the highest mean value, 'b' is the second highest mean value, and so on.

Equal letters imply treatments with means that are statistically equivalent to each other. Additionally, an interaction chart was developed, and the interaction between factors is significant if the lines cross. If no crossover occurs, there was no significance in this interaction, i.e., no reversal of behavior occurred when the factors were combined.

## 3. Results and Discussion

### 3.1. Physical Evaluations

#### 3.1.1. Specific Apparent Mass

Table 3 shows the mean values and the standard deviation of the specific apparent mass for each condition, being the reference condition for non-treated plywoods, and CCA for plywoods produced using veneers treated with chromated-copper-arsenate preservative. There was no significant difference in specific apparent mass between treatments (F-value = 0.4201; $p$-value = 0.5243 > 5%). As for the adhesives, the analysis of variance indicated a significant difference (F-value = 32.7771; $p$-value = $1.33 \times 10^{-5}$ < 5%). As there was an interaction between the adhesive and treatment (F-value = 26.8845; $p$-value = $4.449 \times 10^{-5}$ < 5%), the interaction graph presented in Figure 4a was analyzed.

**Table 3.** Results: specific apparent mass.

| | Specific Apparent Mass (g/cm$^3$) | | |
| --- | --- | --- | --- |
| **Adhesive** | **Preservative Treatment** | | **Mean Values** |
| | **Reference** | **CCA** | |
| Phenol-formaldehyde (PF) | 0.47 (0.03) [1] | 0.57 (0.04) | 0.53 a [2] |
| Castor oil polyurethane (PU) | 0.65 (0.05) | 0.58 (0.03) | 0.62 b |
| **Mean Values** | 0.57 a | 0.58 a | - |

[1] Standard deviation is in parentheses. [2] Same letters are not significantly different (Tukey, $\alpha$ = 5%).

Without the CCA treatment, the specific apparent mass was higher for the PU resin (T2). With CCA, the values were close for both resins. In addition, the veneers treatment led to an increase in the specific apparent mass for phenol-formaldehyde resin (PF) and there was a decrease for castor oil-based polyurethane (PU). The higher apparent density for the PU resin (T2 and T4) may be caused by the higher solids resin content. The decrease after treatment involves the interaction between the components of resin and the CCA present on the veneer. Regarding PF resin, the increase with treatment is evidenced by the impregnation of the treatment components for subsequent production.

The ABIMCI (Brazilian Association of Mechanically Processed Wood Industry) has defined ideal values for specific mass of Brazilian Pine plywood with five layers between 0.49 and 0.57 g/cm$^3$ [32]. Thus, only T1 did not meet the minimum requirement.

There was also no significant difference in the specific apparent mass between the control panels and with CCA in the paper of Ferreira et al. [9] and Mendes et al. [33], nor between the *Pinus sylvestris* panels made with PF and PU by Wilczak et al. [34], which is in line with the results obtained in this research. In the study by Setter et al. [35], the use of distinct formaldehyde-based adhesives (phenol and urea) also did not significantly affect the specific mass of the panels produced with pine veneers.

From the interaction graphics (Figure 4), there was no crossover between the lines. Therefore, it can be concluded that there was no significance in the interaction of the factors, i.e., no reversal of behavior occurred when the factors were combined.

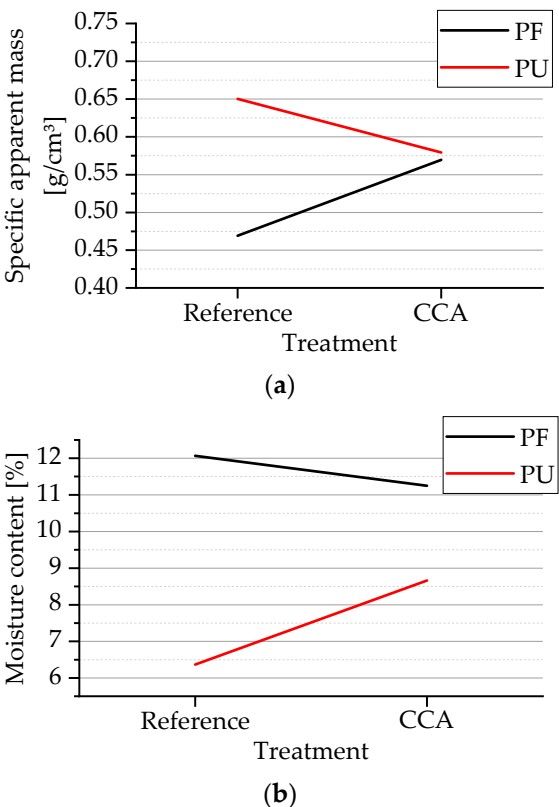

**Figure 4.** Interaction graphics: (**a**) specific apparent mass and (**b**) moisture content.

### 3.1.2. Moisture Content

The average values and the standard deviation of the moisture content are shown in Table 4. There was a significant difference between the types of treatments (F-value = 46.322; $p$-value = $1.276 \times 10^{-6} < 5\%$), as well as between adhesives (F-value = 1467.842; $p$-value = $2.2 \times 10^{-16} < 5\%$). The interaction graphic between adhesives and treatments (F-value = 191.585; $p$-value = $1.049 \times 10^{-11} < 5\%$) is shown in Figure 4b.

**Table 4.** Results: moisture content.

| Adhesive | Moisture Content (%) | | Mean Values |
| --- | --- | --- | --- |
| | **Preservative Treatment** | | |
| | **Reference** | **CCA** | |
| Phenol-formaldehyde (PF) | 12.1 (0.0027) [1] | 11.3 (0.0015) | 11.7 a [2] |
| Castor oil polyurethane (PU) | 6.4 (0.0044) | 8.7 (0.0008) | 7.5 b |
| **Mean Values** | 9.2 a | 10.0 b | - |

[1] Standard deviation is in parentheses. [2] Same letters are not significantly different (Tukey, $\alpha$ = 5%).

As plywood is commonly used as a building material for outdoor applications, understanding its behavior in relation to moisture is extremely important. The moisture absorbed by plywood can compromise its mechanical and physical properties [35,36]. Similar to the paper of Setter et al. [35], the results reached values close to the range of $10 \pm 2\%$, as suggested by ABIMCI [32] e ABNT [37]. It is verified that the moisture content increased with the application of CCA for the PU resin, while for the PF adhesive, there was a decrease in the same condition. The moisture content for the panels produced with PU (T2 and T4) was lower compared to those produced with PF (T1 and T3).

This is in agreement with the results of the study by De Windt et al. [36], where the type of glue influenced the moisture content of uncoated plywood, with residual moisture

being higher in the PF type panels compared to the panels with urea fortified melamine formaldehyde adhesive.

### 3.1.3. Thickness Swelling

A significant difference was found between treatments (F-value = 7.5449; *p*-value = 0.025183 < 5%), as with the adhesives (F-value = 22.8658; *p*-value = 0.001387 < 5%) (Table 5).

**Table 5.** Results: thickness swelling.

| Adhesive | Thickness Swelling (%) | | Mean Values |
|---|---|---|---|
| | **Preservative Treatment** | | |
| | **Reference** | **CCA** | |
| Phenol-formaldehyde (PF) | 6.85 (0.55) [1] | 6.83 (0.27) | 6.84 a [2] |
| Castor oil polyurethane (PU) | 6.13 (0.86) | 4.22 (0.59) | 5.17 b |
| **Mean Values** | 6.49 a | 5.52 b | - |

[1] Standard deviation is in parentheses. [2] Same letters are not significantly different (Tukey, $\alpha$ = 5%).

Furthermore, the treatment and adhesive interaction graphic (F-value = 7.3364; *p*-value = 0.026717 < 5%) is verified in Figure 5a.

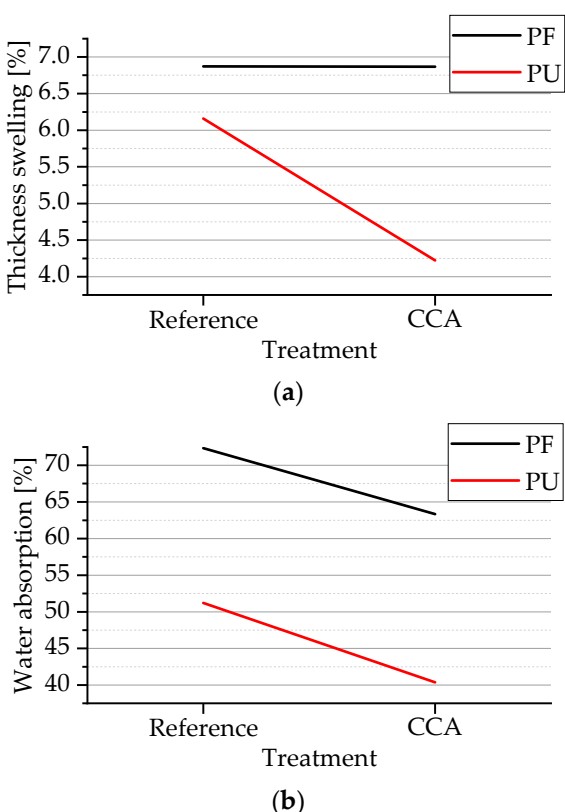

**Figure 5.** Interaction graphics: (**a**) thickness swelling and (**b**) water absorption.

PU panels (T2 and T4) had less thickness swelling than PF panels (T1 and T3) as expected, since the polyurethane resin has a more hydrophobic performance [38]. For PF, there was no difference between treatments, but for PU, the swelling decreased with the treatment. The decrease in thickness swelling for the panels treated with CCA and produced with polyurethane can be explained by the fact that it is a non-polar component. This means that the adhesive does not react with polar components, e.g., water, and its combination

with the CCA-treated veneer—which makes the surface of the veneer hydrophobic—can further inhibit swelling in the thickness.

In the works by Buzo et al. [39] and Sugahara et al. [40], lower thickness swelling values were also found for particle boards made of PU compared to formaldehyde-based resins. It is also observed that the mean values of swelling in thickness, obtained for all treatments of the present study, are a little below the mean value (7.75%) found in the literature for plywood of *Pinus taeda* [33].

From the interaction (Figure 5), there was no crossover between the lines. Therefore, it can be concluded that there was no significance in the interaction of the factors.

### 3.1.4. Water Absorption

For water absorption, there was no significant difference between treatments (F-value = 1.885; $p$-value = 0.6667 > 5%).

As for the adhesives, the test shows that there was a significant difference between them (F-value = 151.1656; $p$-value = $1.896 \times 10^{-14}$ < 5%), as indicated in Table 6. The test also showed interaction between the adhesive and the treatments applied (F-value = 31.1210; $p$-value = $2.543 \times 10^{-6}$ < 5%), according to Figure 5b.

**Table 6.** Results: water absorption.

| | Water Absorption (%) | | |
|---|---|---|---|
| **Adhesive** | **Preservative Treatment** | | **Mean Values** |
| | **Reference** | **CCA** | |
| Phenol-formaldehyde (PF) | 72.8 (0.06) [1] | 63.4 (0.07) | 68.1 a [2] |
| Castor oil polyurethane (PU) | 51.2 (0.06) | 40.3 (0.02) | 45.8 b |
| **Mean Values** | 62.0 a | 51.9 a | - |

[1] Standard deviation is in parentheses. [2] Same letters are not significantly different (Tukey, $\alpha$ = 5%).

Treatment with CCA decreased water absorption for both adhesives. This is justified by the filling with salts of the existing voids in the wood [9]. The PU resin obtained lower absorption than PF in both treatments (reference and CCA). The decrease in absorption for the panels produced with the two adhesives when the treatment with CCA may occur due to the hydrophobic feature acquired by the veneer when the treatment is applied.

The fact that the polyurethane resin has a lower average value can also be explained by its non-polar structure being resistant to water absorption. In the study by Setter et al. [35], the use of different resins (phenol- and urea-formaldehyde) also showed a visible difference for the water absorption. According to the authors, PF adhesive is traditionally used in panels intended for outdoor use, due to its excellent resistance to moisture. Here, the results of the PF adhesive against water absorption were surpassed by the PU.

### 3.2. Mechanical Evaluations

### 3.2.1. Glue Line Shear

There was no significant difference between treatments (F-value = 0.4648; $p$-value = 0.50321 > 5%), nor between the adhesives (F-value = 2.2099; $p$-value = 0.15272 > 5%), as shown in Table 7. There was an interaction between the treatment and the adhesive (F-value = 6.3045; $p$-value = 0.02075 < 5%), as visualized in the graphic of Figure 6.

The treatment did not change the shear strength at the glue line for both adhesives and treatments (Figure 6). The highest value found was for the panel treated with CCA and produced with PU (T4). With the increase in shear strength in the glue line presented by PU and the decrease presented by PF in the presence of CCA, it can be inferred that there was a better interaction between the components that present hydrophobic characteristics. In relation to the phenol-formaldehyde, which is hydrophilic, the interaction decreased the mechanical performance of the panel.

**Table 7.** Results: glue line shear.

| Adhesive | Glue Line Shear (MPa) | | Mean Values |
| | Preservative Treatment | | |
| | Reference | CCA | |
| --- | --- | --- | --- |
| Phenol-formaldehyde (PF) | 2.87 (1.96) [1] | 1.95 (0.54) | 2.41 a [2] |
| Castor oil polyurethane (PU) | 2.35 (0.96) | 3.95 (0.98) | 3.15 a |
| **Mean Values** | 2.61 a | 2.95 a | - |

[1] Standard deviation is in parentheses. [2] Same letters are not significantly different (Tukey, $\alpha = 5\%$).

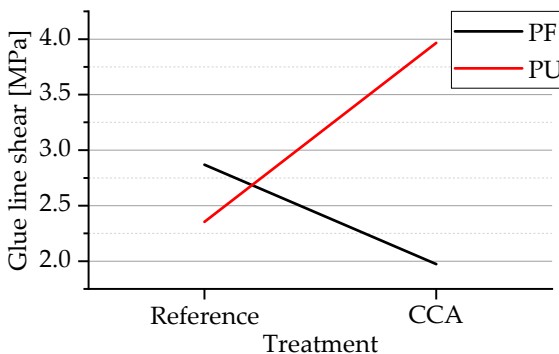

**Figure 6.** Interaction graphics of glue line shear.

The ABNT NBR ISO 12466-2:2012 standard [41] establishes, for panels with shear strength between 0.2 and 0.4 MPa, that the failure in the wood should be higher than 80%. When the resistance is between 0.4 and 0.6 MPa, there is 60% failure; in case of resistance between 0.6 and 1.0 MPa, there is failure greater than 40%; and in panels whose shear strength is higher than 1 MPa, there is no requirement for failure in the wood. So, it is possible to observe that plywood panels produced with both adhesives for all treatments meet the requirements of ABNT NBR ISO 12466-2:2012 and also exceed the minimum of 18 kgf/cm$^2$ recommended by ABIMCI [32] for five layer panels. Wilczak et al. [34] also obtained values above 2 MPa for plywood panels produced with PF and PU, and their reference panels. Furthermore, Setter et al. [35] reached 1.68 MPa for pine and PF resin panels.

From the interaction graphics (Figure 6), it can be observed that there was crossover between the lines. Therefore, it can be concluded that there was significance in the interaction of the factors, i.e., there was an inversion of values between the factors.

### 3.2.2. Parallel and Perpendicular Static Bending

In Table 8, we can observe that there was a significant difference between the treatments (F-value = 34.9467; $p$-value= 0.0003571 < 5%) and among the adhesives (F-value = 18.0815; $p$-value = 0.0027907 < 5%) for the modulus of elasticity in the parallel direction (MOE∥) and there was no interaction between the treatments and adhesives (F-value = 1.7636; $p$-value = 0.2208123 > 5%), according to Figure 7a. For the modulus of rupture in the parallel direction (MOR∥), there was no significant difference between treatments (F-value = 0.9314; $p$-value = 0.36276 > 5%), nor among the adhesives used (F-value = 4.3061; $p$-value = 0.07166 > 5%). In addition, there was an interaction between the treatment and resins (F-value = 9.2684; $p$-value = 0.01596 < 5%), according to Figure 7b. For both types of treatments, the PF adhesive obtained higher values of MOE∥. Higher values of MOE∥ were also found for panels with CCA. For MOR∥, the reference treatment PF (T1) had the highest value. The MOE∥ the treatment directly interferes because for the two adhesives, there is an increase in the modulus with the addition of the CCA. For this property, the interaction between adhesives and the treated veneers contributed to an increase in the medium value.

**Table 8.** Parallel static bending.

| Adhesive | MOE∥ [3] (MPa) | | Mean Values |
|---|---|---|---|
| | **Preservative Treatment** | | |
| | **Reference** | **CCA** | |
| PF | 4758.58 (417.17) [1] | 6711.09 (325.42) | 5734.83 a [2] |
| PU | 3969.92 (767.00) | 5206.10 (68.10) | 4588.01 b |
| **Mean Values** | 4364.25 a | 5958.59 b | - |

| Adhesive | MOR∥ (MPa) | | Mean Values |
|---|---|---|---|
| | **Preservative Treatment** | | |
| | **Reference** | **CCA** | |
| PF | 46.92 (8.92) [1] | 40.32 (2.47) | 43.62 a [2] |
| PU | 30.67 (3.16) | 43.40 (4.88) | 37.03 a |
| **Mean Values** | 38.79 a | 41.86 a | - |

[1] Standard deviation is in parentheses. [2] Same letters are not significantly different (Tukey, $\alpha = 5\%$); [3] modulus of rupture in the parallel direction.

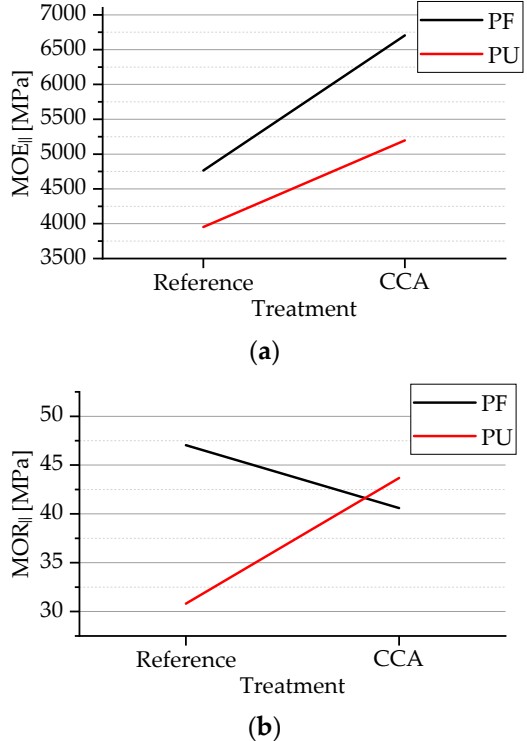

**Figure 7.** Interaction graphics: parallel static bending (**a**) MOE∥ and (**b**) MOR∥.

As for MOR∥, it is noteworthy that phenol-formaldehyde without the application of treatment has better interactions with the veneer than the polyurethane resin. Such behavior can be observed because the adhesive is hydrophilic and reacts better with the moisture in the veneer without treatment.

From the interaction graphics (Figure 7), there was no crossover between the lines in MOE∥ (Figure 7a) and there was a crossover in MOR∥ (Figure 7b). Therefore, it can be concluded that there was no significance in the interaction of the factors for MOE∥, but there was significance for MOR∥, leading to an inversion of values between the factors.

The ABIMCI [32] defines 4782.51 MPa as the minimum value for MOE∥. Thus, only the T2 treatment did not meet the requirements. The low point value obtained for the panels produced with untreated PU resin (T2) can be related to the quality of the veneers.

As already verified in a study carried out by Kazmierczak et al. [42], the quality of the veneer, as well as the wood species, interfere with the plywood properties. Thus, the low results may have been influenced by the presence of defects in the panel veneers. For MOR‖, the recommended minimum is 22.95 MPa, a value reached by all the treatments.

In their research, Wilczak et al. [34] reached 61.19 MPa (PF) and 63.32 MPa (PU) for MOR‖ and 7190 MPa (PF) and 6610 MPa (PU) for MOE‖. Setter et al. [35] obtained 38.92 MPa and 5520 MPa for MOR and MOE ‖, respectively, in panels with PF resin. Mendes et al. [33] reached 44.2 MPa (MOR‖) and 7071 MPa (MOE‖) in panels manufactured with CCA and 48.48 MPa (MOR‖) and 7924 MPa (MOE‖) in reference panels (both with PF).

For static bending in the perpendicular direction (Table 9), there was no significant difference for the modulus of elasticity (MOE⊥) of the treatments (F-value = 0.9051; *p*-value = 0.3693 > 5%), nor for the MOE⊥ of the different adhesives (F-value = 1.1901; *p*-value = 0.36276 > 5%). There was no interaction (Figure 8a) for MOE⊥ (F-value = 0.0047; *p*- value = 0.9471 > 5%). For modulus of rupture in the perpendicular direction (MOR⊥), there was a significant difference between treatments (F-value = 6.3139; *p*-value= 0.03622 < 5%), which did not happen for the adhesives (F-value = 0.0355; *p*-value = 0.85518 > 5%). For MOR⊥ (Figure 8b), there was no interaction between the treatments and adhesives (F-value = 0.1517; *p*-value = 0.70710 > 5%).

**Table 9.** Perpendicular static bending.

| | MOE⊥ [3] (MPa) | | |
|---|---|---|---|
| **Adhesive** | **Preservative Treatment** | | **Mean Values** |
| | **Reference** | **CCA** | |
| PF | 1648.34 (395.51) [1] | 1858.20 (579.84) | 1753.27 a [2] |
| PU | 1372.75 (347.64) | 1615.17 (253.98) | 1493.96 a |
| **Mean Values** | 1510.54 a | 1736.68 a | - |
| | **MOR⊥ (MPa)** | | |
| **Adhesive** | **Preservative Treatment** | | **Mean Values** |
| | **Reference** | **CCA** | |
| PF | 18.95 (0.97) [1] | 23.49 (5.89) | 21.22 a [2] |
| PU | 18.52 (3.00) | 24.73 (3.21) | 21.62 a |
| **Mean Values** | 18.73 a | 24.11 b | - |

[1] Standard deviation is in parentheses. [2] Same letters are not significantly different (Tukey, $\alpha$ = 5%). [3] Modulus of rupture in the perpendicular direction.

Despite the mean values for MOE⊥ of PF being higher compared to PU, there were no significant differences. It is noteworthy that the hydrophobic aspect of the resin can interfere with its penetration and, consequently, with mechanical properties [38]. Thus, treatment with CCA also did not significantly alter the mean value of MOE⊥. It can be observed that there was an increase in MOR⊥ for both resins when the CCA treatment was applied to the panel veneers. There was no significant difference between either of the resins.

For the MOE⊥, it can be inferred that the factors involved do not change the mechanical property, despite a small increase in the mean value with the application of the treatment. As for MOR⊥, an increase in the average value was found with the CCA treatment, both for PF and PU. This effect can possibly be attributed to the better interactions of the adhesives in relation to the treated veneer.

The ABIMCI [32] defines 1866.79 MPa as the minimum value for MOE⊥; thus, no treatment met the requirements. Due to the fact that the veneers used in this research were donated by a domestic company, most of the wooden veneers applied to the panels manufacture came from classes II, III and IV according to the classification of ABNT NBR ISO 2426-3 [25], containing some defects. In this way, when assembling the panels, an attempt was made to place the best quality veneers in the external layers of the panel.

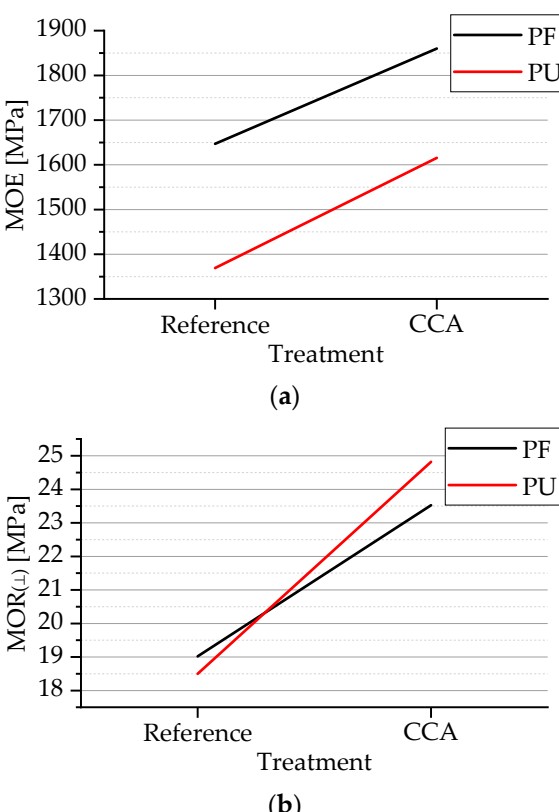

**Figure 8.** Interaction graphics: parallel static bending (**a**) MOE⊥ and (**b**) MOR⊥.

However, some panels were produced only with class III and IV veneers, which are the ones with the highest incidence of defects, such as knots and reverse grains. For MOR⊥, the minimum recommended is 15.49 MPa, whose value was achieved by all the proposed treatments.

Wilczak et al. [34] reached 32.18 MPa (PF) and 34.78 MPa (PU) for MOR⊥ and 2250 MPa (PF) and 2130 MPa (PU) for MOE⊥. Setter et al. [35] obtained 23.26 MPa and 4240 MPa for MOR and MOE⊥ for panels made with PF resin, respectively. In addition, Mendes et al. [33] reached 28.74 MPa and 1662.78 MPa for MOR and MOE⊥ in panels made with CCA and 33.66 MPa and 2243.2 MPa for MOR and MOE⊥ in reference panels (both produced with PF resin).

From the interaction graphics (Figure 8), it can be observed that there was no crossover between the lines in MOE⊥ (Figure 8a) and there was a crossover in MOR⊥ (Figure 8b). Therefore, it can be concluded that there was no significance in the interaction of the factors for MOE⊥, but there was significance for MOR⊥, which caused an inversion of values between the factors.

## 4. Conclusions

The present study showed important results in relation to plywood, which include the following conclusions:

- The preservative treatment with CCA was usually efficient, as the same treatment improved the properties, above all, when compared to the reference panels.
- In the panels produced with PU, lower moisture content and better interactions with water (less swelling in thickness and water absorption) were observed.
- Most treatments had increases in the modulus of elasticity and modulus of rupture for both adhesives when the CCA treatment was applied to the wood veneers.
- Comparing both adhesives, PF presents the better values of modulus of elasticity.
- Glue line shear for all treatments met the requirements defined by the ABNT NBR ISO 12466-2: 2012.

- In the scientific scope, there were not significant limitations in the present research. Under commercial perspectives, the main limitation of this study may be related to the possible adaptation of the existing industrial plants oriented to the manufacture of PF-based plywoods with regard to the insertion of new processes driven by PU resin. However, this change will be in charge of the consideration of PU resin by the plywood industry, since it will imply adjustments in the manufacturing parameters, such as resin viscosity, mat pressing, pressing temperature, etc. Another limitation may be attributed to the utilization of the CCA wood preservative, which has been restricted and/or prohibited in some markets of North America and Europe.
- Despite the commercial uses of CCA-treated PF-based plywoods, the utilization of PU resin in plywood production represented a tangible novelty for the timber industry, above all, to satisfy severe conditions of wood uses in the Southern Hemisphere.

**Author Contributions:** Conceptualization, E.S., B.C. and H.S.; methodology, E.S. and C.C.; software, E.S. and H.S.; validation, E.S. and H.S.; formal analysis, E.S. and H.S.; investigation, E.S. and E.F.; resources, E.F., H.S. and B.C.; data curation, E.S. and B.C.; writing—original draft preparation, E.S., F.A. and V.D.A.; writing—review and editing, E.S., F.A., V.D.A. and C.C.; visualization, E.S., F.A., V.D.A., E.F. and H.S.; supervision, A.C. and C.C.; project administration, A.C. and C.C. All authors have read and agreed to the published version of the manuscript.

**Funding:** This study was financed by the Pró-Reitoria de Pesquisa, Inovação e Pós-Graduação of Instituto Federal de Rondônia (PROPESP/IFRO), FAPESP (grant #2015/04660-0) and the Coordenação de Aperfeiçoamento de Pessoal de Nível Superior, Brasil (CAPES), Finance Code 001.

**Data Availability Statement:** The data presented in this study are available upon request from the corresponding author.

**Acknowledgments:** We would like to acknowledge the Pró-Reitoria de Pesquisa, Inovação e Pós-Graduação of Instituto Federal de Rondônia (PROPESP/IFRO) and the Coordenação de Aperfeiçoamento de Pessoal de Nível Superior, Brazil (CAPES).

**Conflicts of Interest:** The authors declare no conflict of interest.

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
