# Peer review of "Comparative Study of Plywood Boards Produced with Castor Oil-Based Polyurethane and Phenol-Formaldehyde Using Pinus taeda L. Veneers Treated with Chromated Copper Arsenate"

_forests, doi:10.3390/f13071144_

Round 1
Reviewer 1 Report
The authors have addressed all my previous comments/remarks. I believe the revised version of the manuscript has been improved and can be accepted for publication.
Author Response
R: We appreciate your efforts to support us in this revision process, which is being essential to improve our paper quality.
Reviewer 2 Report
The review report, manuscript „Comparative Study of CCA Treated PlywoodProduced with Phenol-formaldehyde and Castor Oil-based Polyurethane“.The manuscript is well written, some minor revisions are needed beforepublication.The Title: I suggest not using abbreviations in the Title.The abstract is well prepared and contains the main finding of theresearch.The Introduction:Line 31-32: Please focus more on plywood, as it is the topic of yourmanuscript.Line 40-41: Please find a more appropriate reference to this statement.Lines 49-50: This is well-known information, no need to explain.Lines 52-53: This is not completely true, please revise. Also, it is awood-based panel, not a wood panel.Lines 59: Please discuss also other methods of impregnation treatment.Please explain why did you use CCA in your research.Lines 61-99: This part is well prepared.Please check the whole manuscript for abbreviations. Some are explainedmore than once (PF, CCA, etc.).The Material and Methods, Results, and discussion parts are well done,results are clearly presented, detailed, and properly discussed withrelevant research works.The Conclusions are also well written.Please add limitations of the research and implications for furtherresearch.
Author Response
The review report, manuscript „Comparative Study of CCATreated Plywood Produced with Phenol-formaldehyde and Castor Oil-based Polyurethane“. The manuscript is well written, some minor revisions are needed before publication.
R: We appreciate your efforts to support us in this revision process, which is being essential to improve our paper quality.
The Title: I suggest not using abbreviations in the Title.
R: Title was rewritten to specify the CCA term.
The abstract is well prepared and contains the main finding of the research.
R: Thank you for your positive opinion about the abstract.
The Introduction:
Line 31-32: Please focus more on plywood, as it is the topic of your manuscript.
R: Usually, first paragraph are used to describe a global scenario, as we did. As you can check, the first paragraph approaches the wood as a resource for industry, above all, as a silvicultural alternative in the current moment to supply industry and markets. Thus, the specific focus on plywood is completely unnecessary in this first paragraph, as we are sequentially approaching on plywoods from the second paragraph to fourth paragraph. But, we inserted in this part more sentences about plywood to satisfy the reviewer’s expectation.
Line 40-41: Please find a more appropriate reference to this statement.
R: we rearranged this statement, exemplifying in sequence the current global scenario in the consumption and production.
Lines 49-50: This is well-known information, no need to explain.
R: we deleted this paragraph, although we inserted another sentence to start the approach of plywoods.
Lines 52-53: This is not completely true, please revise. Also, it is a wood-based panel, not a wood panel.
R: we corrected the sentence and revised the term.
Lines 59: Please discuss also other methods of impregnation treatment.
R: using the reviewer’s suggestion to focus more on the main topics, we imagine the discussion about other methods is unnecessary.
Please explain why did you use CCA in your research.
R: This information was clearly approached and justified in the objective. Please, check our arguments, as we stated our considerations to each material. About CCA, we stated that: “CCA preservative was considered due to the greater aggressiveness of this chemical compound in protecting against wood degradation. Despite some restrictions and prohibitions on the use of CCA in different countries of the Northern hemisphere, the warmer climate and the greater proliferation of wood-decaying insects in the Southern hemisphere region represent complex obstacles that still justify the use of this more powerful preservative to preserve wood-based parts and panels”.
Lines 61-99: This part is well prepared.
R: Thank you for your positive opinion about the abstract.
Please check the whole manuscript for abbreviations. Some are explained more than once (PF, CCA, etc.).
R: As tables contain the explanation for PF and PU resins, and CCA, the insertion of both terms will establish an intense repetition of words. Even so, we inserted the complete name of resins in the initial part of results and other two points as a strategy to satisfy your suggestion.
The Material and Methods, Results, and discussion parts are well done, results are clearly presented, detailed, and properly discussed with relevant research works.
R: Thank you for your positive opinion about the abstract.
The Conclusions are also well written.
R: Thank you for your positive opinion about the conclusion.
Please add limitations of the research and implications for further research.
R: we inserted an initial sentence in the penultimate paragraph of conclusions to declare that no limitations were observed in the present research, as implications may be observed in some cases as described only for commercial view. Thus, we described the possible limitations and implications for this case in sequence.
This manuscript is a resubmission of an earlier submission. The following is a list of the peer review reports and author responses from that submission.
Round 1
Reviewer 1 Report
The manuscript deals with the investigation and evaluation of the effect of chemical treatment with chromated-copper-arsenate on the physical and mechanical properties of plywood panels, bonded with phenol-formaldehyde and castor oil-based polyurethane adhesives. In general, the manuscript is well-written and structured, but needs some minor improvements before acceptance for publication in the Forests Journal. Please, see below my comments on your work:
In general, the title (lines 2-3), the abstract (lines 13 to 26) and the keywords (lines 28-29) correspond to the aims and objectives of the manuscript. I would recommend to revise the title a bit in order to be more specific about the types of adhesives used.
In general, the abstract is informative and contains the main findings of the article. However, I would recommend to revise it by adding some specific results obtained from your study. For example, the first two sentences are well-known facts and are too general, I’d recommend to delete/revise them.
Line 17: “plywood options” – what do you mean by that? Two types of plywood? Please revise.
Lines 28-29: I’d recommend to change “wooden composites” with “wood-based composites”, and “physical and mechanical evaluations” with “physical and mechanical properties” in order to be more accurate.
Lines 60-61: The statement “However, due the concerns about the potential indoor pollution, the development of environmentally friendly adhesives that are nontoxic and renewable is meaningful.” is general true, but here I’d recommend to extend it a bit, stating the main health hazard related to emission of VOCs and free formaldehyde from the finished wood composites. Please check this relevant reference here: https://doi.org/10.1080/17480272.2022.2056080
Lines 65-66: There are many successful examples of producing plywood panels with bio-based adhesives, please check this relevant references:
https://doi.org/10.1021/acsami.2c02859
https://doi.org/10.3390/polym14102111
https://doi.org/10.1007/s10570-022-04611-9
Line 80: the botanical name of the paper mulberry species should be written like this: Broussonetia papyrifera, please revise.
Line 97: please explain why did you select loblolly pine (Pinus taeda) for your research.
In general, the Introduction part is well-written and informative, and provides relevant background of the research. The inclusion of additional references, as recommended, will increase the scientific soundness of the work.
Line 103: the dimensions of veneer sheets should be given like this: 400 mm × 400 mm × 2.3 mm or 400 × 400 × 2.3 mm3, please revise. In addition to veneer dimensions, please provide info about their moisture content.
Line 125: please explain why did you use mmHg?
Lines 148-151: please explain/justify the selected hot pressing regime parameters, i.e. temperature, pressing pressure and press time.
Line 154: “Were performed physical and mechanical evaluations…” it is not grammatically correct to start a sentence like this. A general remark: please improve the language and style of the whole manuscript. Preferably, this should be made by a native speaker.
Line 167: please provide relevant information on the software product used (company, city, country).
Overall, the Materials and Methods section is well written but should be further elaborated based on the comments above.
In general, the results of the study are detailed, informative and discussed with relevant research works in the field.
The Conclusion part (lines 416-426) reflects the main findings of the manuscript. Please add the limitations of your research as well as potential use of your results in the industrial practice.
The References cited are appropriate to the topic of the manuscript. Inclusion of additional references, especially in the Introduction section, will significantly increase the scientific merit of the presented manuscript.
Best regards!
Reviewer 2 Report
This is an old topic. CCA treated wood had been prohibited for application in many countries as a result of environmental pollution.
Reviewer 3 Report
It is unclear how this work is novel. CCA treated plywood has been commercialized for many decades. There are many studies examining treatment, bondability, and mechanical properties - mostly from the 20th century. A more thorough literature review is needed. The oldest work cited in this manuscript was from 2006.
The interaction graphs are unnecessary as they duplicate data already presented in the tables.
Table 8 - check significant figures